# Emerging Role of PYHIN Proteins as Antiviral Restriction Factors

**DOI:** 10.3390/v12121464

**Published:** 2020-12-18

**Authors:** Matteo Bosso, Frank Kirchhoff

**Affiliations:** Institute of Molecular Virology, Ulm University Medical Center, 89081 Ulm, Germany; matteo.bosso@uni-ulm.de

**Keywords:** PYHIN, DNA sensing, restriction factors, viral counteraction, immune evasion

## Abstract

Innate immune sensors and restriction factors are cellular proteins that synergize to build an effective first line of defense against viral infections. Innate sensors are usually constitutively expressed and capable of detecting pathogen-associated molecular patterns (PAMPs) via specific pattern recognition receptors (PRRs) to stimulate the immune response. Restriction factors are frequently upregulated by interferons (IFNs) and may inhibit viral pathogens at essentially any stage of their replication cycle. Members of the Pyrin and hematopoietic interferon-inducible nuclear (HIN) domain (PYHIN) family have initially been recognized as important sensors of foreign nucleic acids and activators of the inflammasome and the IFN response. Accumulating evidence shows, however, that at least three of the four members of the human PYHIN family restrict viral pathogens independently of viral sensing and innate immune activation. In this review, we provide an overview on the role of human PYHIN proteins in the innate antiviral immune defense and on viral countermeasures.

## 1. Introduction

Viruses strictly rely on their host cells for replication and spread. However, although viral pathogens are capable of exploiting numerous cellular factors and pathways, the cell does not provide a friendly environment. As a consequence of countless past encounters with viral pathogens, mammalian cells have evolved sensors of foreign invaders that alert and activate a large variety of antiviral effector proteins [1,2,3,4,5]. Pyrin and hematopoietic interferon-inducible nuclear (HIN) domain-containing (PYHIN) protein family members have initially been recognized as novel types of pattern recognition receptors (PRRs) and proposed to trigger innate immune responses and inflammasome activation upon detection of pathogen-derived DNA [6,7,8]. However, most of the evidence comes from numerous studies on the PYHIN protein AIM2 (Absent In Melanoma 2), a cytoplasmic sensor of double-stranded DNAs [9,10,11,12]. In contrast to AIM2, however, the remaining human PYHIN proteins are predominantly localized in the nucleus and accumulating evidence suggests that they exert antiviral effects by suppressing viral transcription rather than by sensing viral DNAs. This review aims at summarizing recent findings that support a role of human PYHIN proteins as antiviral restriction factors.

## 2. The PYHIN Protein Family

PYHIN proteins are characterized by two functional domains: an N-terminal pyrin domain (PYD) and at least one C-terminal hematopoietic interferon-inducible nuclear protein with a 200-amino-acid repeat domain (HIN200) (Figure 1). The PYD is part of the bigger superfamily of death domains (DD) characterized by an alpha-helical-based folding, promoting homo- or hetero-typic interactions with other PYD-containing proteins. PYD-PYD interactions regulate a variety of cellular processes, ranging from inflammation and immunity to apoptosis and cell cycle [13]. The HIN domain is only found in PYHIN family members and promotes DNA-binding in a non-sequence specific fashion via tandem oligonucleotide/oligosaccharide-binding (OB) folds [14,15]. Sequence independent DNA binding is achieved by electrostatic interactions between specific side chains of positively charged HIN domain amino acid residues and the phosphate groups in the DNA backbone [16]. HIN domains have been classified in three subfamilies, designated -A, -B, and -C, based on the amino acidic sequence following a conserved MFHATVAT motif [17].

*PYHIN* coding genes are exclusively found in mammals and their numbers range from 1 in horses to up to 13 in mice [18,19]. Humans encode four PYHIN proteins (Figure 1): γ-IFN-Inducible protein 16 (IFI16), IFN-Inducible protein X (IFIX) also known as Pyrin and HIN domain-containing protein 1 (PYHIN1), Myeloid Nuclear Differentiation Antigen (MNDA), and Absent In Melanoma 2 (AIM2) [7]. MNDA was the first human PYHIN family member that was discovered and is found in the nucleus of myeloid cells [20]. The differentiation stage- and lineage-specific expression of MNDA [21] suggested a potential role in transcriptional regulation of myeloid cell differentiation. In this context, MNDA has been shown to interact with nucleolin and nucleophosmin [22,23], as well as to bind and enhance DNA-binding affinity of the transcription factor Ying Yang 1 (YY1) [22]. Furthermore, MNDA itself has been proposed to directly regulate transcription in monocytes [24].

The second PYHIN family member to be discovered was IFI16 [25]. IFI16 is expressed in various cell types and tissues. Initial studies focused on the role of IFI16 in modulating transcription and the cell cycle. IFI16 interacts with the tumor suppressor p53 [26] and increases its transcriptional activity [27]. More recent findings indicate that IFI16 is required for optimal RNA polymerase type II (RNA pol II) binding to the promoters of interferon (IFN)-α and IFN-stimulated genes (ISGs) [28]. In addition, it has been shown that IFI16 acts as a transcriptional repressor that inhibits a minimal promoter containing an intact Sp1 binding site [29,30]. These early studies suggested that the inhibitory activity is dependent on the HIN domains [29] and that IFI16 may compete with Sp1 for DNA binding [30]. IFI16 was also shown to directly interact with the p53 and c-myc gene promoters [31] and to promote cell cycle arrest and induction of cellular senescence [32]. Finally, it has been reported that IFI16 interacts with the breast cancer type-1 susceptibility protein (BRCA1) and promotes p53-mediated apoptosis [33].

Although AIM2 is the best characterized member of the human PYHIN family, it was only the third to be discovered, initially as being suppressed in a malignant melanoma cell line [34]. Numerous studies have examined and established its role as a cytosolic DNA sensor. In comparison, the role of AIM2 in cancer development is less clear. AIM2 has long been regarded as a tumor suppressor gene because reduced expression has been reported for colon, liver, breast, and liver cancers [10,35]. More recent studies have shown, however, that AIM2 is upregulated in squamous cell carcinoma and lung cancer [36,37]. Thus, further studies are required to elucidate the role of AIM2 in cancer.

The latest member of the PYHIN family to be discovered was IFIX (also named PYHIN1) as a putative tumor suppressor gene in breast cancer [38]. IFIX was shown to exert antiproliferative activity by interacting and destabilizing the p53 E3-ubiquitin ligase HDM2 via its HIN-A domain, leading to increased p53 protein levels and reduced cell proliferation [38]. Additionally, IFIX has been reported to reduce breast cancer cell metastasis by upregulating the metastasis suppressor maspin, potentially by degrading the histone deacetylase 1 (HDAC1) [39].

## 3. PYHIN Proteins as Innate DNA Sensors

The early findings that *PYHIN* coding genes cluster within an IFN-inducible locus [17,40] and that the HIN domain allows DNA binding [14] suggested that PYHIN proteins might play a role in innate immunity. The role of AIM2 as an innate sensor of cytosolic dsDNA has been the topic of recent in-depth reviews [9,10,35,41]. In brief, recent data suggest that AIM2 forms small oligomers that fail to activate downstream signaling in the absence of dsDNA in the cytoplasm (Figure 2a) [42]. Upon binding to cytosolic DNA, AIM2 assembles filamentous structures, whose size seems to depend on the length of the accessible nucleic acids. Assembly of AIM2 oligomers triggers the inflammasome response by nucleating the polymerization of the downstream adaptor apoptosis-associated speck-like protein containing a caspase activation domain (ASC) via PYD-PYD interactions [11,42]. The formation of such filamentous structures generates recruitment platforms for caspase-1, which upon cross-induced proteolytic activation further amplify inflammasome signaling [43]. Activated caspase-1 in turn promotes proteolytic activation of proIL-1β and proIL-18, the precursors of two inflammatory cytokines [11], as well as pyroptotic cell death [44] (Figure 2a). Albeit AIM2 is mostly known to sense bacterial-derived DNA, there is evidence for its involvement in detecting vaccinia virus- and human papillomavirus-derived nucleic acids [45,46].

Independent studies also reported a role of IFI16 in the sensing of pathogen-derived nucleic acids in both the nucleus [47,48,49] and in the cytosol [50,51,52,53,54]. It has been suggested that the better accessibility of foreign DNAs entering into the nucleus compared to the densely packed chromatin-associated cellular DNA allows IFI16 to distinguish between friend and foe [55]. Upon binding to relatively long stretches of accessible nuclear dsDNA, several IFI16 molecules may aggregate and generate clusters that are stabilized by PYD-mediated oligomerization (Figure 2b) [56]. Infection by different viruses has been reported to trigger IFI16 translocation to the cytosol [57,58] and to cooperate with the cyclic guanosine monophosphate-adenosine monophosphate (cGAMP) synthase (cGAS) to enhance the stimulator of interferon gene (STING)-mediated activation of IFN expression [51,59,60]. IFI16 senses multiple herpesvirus-derived DNAs [61] and intermediate products of the reverse transcription process in cells that are abortively infected with HIV-1 [52]. However, its role as an innate DNA sensor is under debate. Recent studies showed that IFI16 knock out does not affect IFN production in infected primary foreskin fibroblasts [62,63]. In addition, it has been reported that IFI16 promotes activation of the cGAS-STING pathway upon sensing of exogenous DNA by cGAS-STING in primary macrophages and skin keratinocytes, rather than directly acting as DNA sensor [59,64]. Altogether, numerous studies reported that PYHIN proteins act as sensors of foreign DNAs in various cell types. Strikingly, however, deletion of the whole *Pyhin* locus in mice encoding for a total of 13 PYHIN proteins did not affect viral DNA sensing and subsequent IFN production in vivo [62], raising the possibility that the major function of PYHIN proteins has yet to be clarified.

## 4. Viral Restriction by PYHIN Proteins Independently of Immune Sensing

Although lack of all 13 *Pyhin* genes in mice did not affect their type I IFN response to viral infection [62], it was associated with substantially higher levels of Friend-Virus viremia during the acute phase of infection [65]. This finding agrees with the accumulating evidence that PYHIN proteins are capable of inhibiting viral pathogens independently of viral immune sensing and IFN induction. Especially IFI16 is emerging as an important antiviral restriction factor capable of suppressing viral transcription of herpes-, retro-, papilloma-, and hepatitis viruses by various non-exclusive mechanisms.

Herpes simplex virus type 1 (HSV-1) is a highly contagious and prevalent virus that establishes latent infections in the infected host [66]. Most infections in healthy individuals are asymptomatic, although periodical viral reactivation may give rise to painful oral or genital blisters or ulcers. In addition, HSV-1 may cause pregnancy loss as well as severe disease in neonates and immunocompromised people. HSV-1 was the first virus shown to be restricted by endogenous IFI16 expression [67]. Early studies reported that IFI16 affects the overall chromatin and transcriptional landscape of viral immediate-early, early, and late gene promoters (Figure 3a). In particular, IFI16 promotes trimethylation of histone H3 lysine 9 (H3K9me3), a heterochromatin mark, and reduces trimethylation of histone H3 lysine 4 (H3K4me3), linked to actively transcribed genomic regions [68,69]. Consequently, IFI16 suppresses binding of host transcription factors required for expression of immediate-early viral genes, as well as RNA pol II [68].

IFI16 was also shown to suppress HSV-1 transcription by supporting the recruitment of nuclear domain 10 (ND10) bodies to viral genomes [70]. ND10 bodies include the promyelocytic leukemia (PML) protein [71,72], the nuclear antigen Sp100 [72], and the death domain-associated protein 6 (Daxx) [71]. The contribution of ND10 bodies to IFI16-mediated suppression of HSV-1 transcription is under debate. It has been reported that knock-out of IFI16 but not PML enhances infectious HSV-1 yield from primary foreskin fibroblasts [63]. IFI16 and Daxx exert synergistic effects on replication of a mutant HSV-1 variant unable to antagonize these antiviral factors [73], suggesting independent mechanisms of action. IFI16 has further been shown to silence transcription of both parental and progeny viral genomes in the absence of a virus-encoded antagonist [73]. This may be achieved by assembling filamentous structures on the viral DNA, which function as recruitment platforms for other epigenetic repressors, such as ATRX, and reduce the binding of elongation-competent RNA pol II (Figure 3a) [74]. While further studies are required to fully elucidate the inhibitory mechanisms, these results suggest that IFI16 may play a role in at least two defense pathways against HSV‑1, i.e., chromatin remodeling and ND10 assembly and recruitment. However, the possibility that both are linked and recruitment of ND10 by IFI16 is required for histone deacetylation can also not be excluded. Finally, the ability of IFI16 to suppress HSV-1 gene expression is shared by IFIX [75], but the exact mechanism remains to be determined.

Kaposi’s sarcoma-associated herpesvirus (KSHV) is an oncogenic virus, known to establish life-long latent infections in healthy individuals and to cause malignancies in immunocompromised patients [76]. IFI16 has initially been shown to bind to the KSHV genome during both de novo and latent infection [77] and might be required for maintaining the virus in a latent state [78]. Recently, it has been reported that IFI16 forms a complex with two H3K9 histone methyltransferases, SUV39H1 and G9a-like protein (GLP) (Figure 3b) [79]. Upon recognition of the viral genome, IFI16 recruits these factors to the viral lytic genes. Subsequent accumulation of H3K9me3 facilitates binding of the heterochromatin protein 1 α (HP1α), a factor involved in formation of repressive hetero-chromatin [80], and might consequently promote silencing of viral gene expression.

Human papillomaviruses (HPVs) are common sexually transmitted DNA viruses that frequently cause asymptomatic infections that are spontaneously resolved. Some strains, however, such as HPV16 and HPV18, are associated with a high risk for cancer and account for about 70% of all cervical carcinoma cases [81]. IFI16 has been shown to reduce viral replication and both early and late gene expression in cells harboring the episomal HPV18 genome [82]. As outlined above for HSV-1 and KSHV, IFI16 may enforce repression of HPV by reducing the levels of H3K4me2 and increasing the binding of H3K9me2 on viral promoters [82]. Notably, IFI16 also inhibits viral transcription in HeLa cells immortalized by the high-risk HPV18 [83], suggesting effects on both the circularized and integrated form of the viral genome. HPV gene expression is regulated by the long control region (LCR), which harbors regulatory sites for genome replication, termination, and polyadenylation of late viral genes as well as binding sites for transcription factors [84]. Amongst them, an Sp1 binding site has been shown to be required for efficient LCR-driven gene expression [85,86]. Notably, IFI16 failed to reduce the activity of an LCR-luciferase reporter construct carrying a mutated Sp1 binding site, suggesting that IFI16 suppresses HPV18 gene expression by Sp1-dependent mechanisms [82].

Hepatitis B virus (HBV) is a hepatotropic virus that remains an important health burden despite the availability of an effective vaccine. It has been estimated that about 3.5% of the world population is chronically infected by HBV and at risk of developing cirrhosis and hepatocellular carcinoma [87]. A recent study reported that IFI16 recognizes the covalently closed circular DNA (cccDNA) of HBV by selectively binding to an IFN-stimulated response element (ISRE; Figure 4) [88]. This element is located in the enhancer 1/X gene promoter region of HBV [89] and renders it susceptible to IFN-α by suppressing viral gene expression via epigenetic modifications [90,91]. IFI16 interaction with the viral ISRE reduces binding of H3K4me3, the acetyltransferase p300/CREB-binding protein (CBP), and the transcription factors STAT1 and STAT2 (Figure 4). Simultaneously, IFI16 promotes recruitment of deacetylases histone deacetylase 1 (HDAC1), sirtuin 1 (Sirt1), and the lysine methyltransferase enhancer of zeste homolog (EZH2) [88]. Induction of IFI16 may contribute to IFN-α-mediated epigenetic silencing of HBV gene expression [88].

Human cytomegalovirus (HCMV) is another very prevalent herpesvirus, causing asymptomatic infections in immunocompetent hosts and a variety of clinical complications during fetal development and in immunocompromised patients [92]. IFI16 has been shown to reduce HCMV mRNAs synthesis. It has been proposed that the inhibitory mechanism may involve sequestration of the transcription factor Sp1 by IFI16, leading to reduced promoter activity of HCMV genes encoding the viral DNA polymerase (UL54) and its subunit UL44 (Figure 5a) [67]. Altogether, these results support the role of IFI16 in restricting various Herpesviruses.

Sp1 is also important for transcription of HIV-1, the etiologic agent of AIDS. IFI16 has initially been identified as a potential inhibitor of HIV-1 by a genome-wide screen for cellular proteins sharing properties of known antiretroviral restriction factors [93]. Subsequent studies revealed that IFI16, as well as IFIX and MNDA, inhibit HIV-1 transcription and replication in CD4^+^ T lymphocytes and/or monocyte-derived macrophages by reducing the availability of the transcription factor Sp1 (Figure 5b) [65,94]. Mutagenesis and functional analyses revealed that the N-terminal pyrin domain and nuclear localization are sufficient for antiretroviral activity and Sp1 binding. The finding that HIN domains thought to be critical for dsDNA sensing by human PYHIN proteins [16,54] are dispensable for antiretroviral activity further supports the idea that PYHIN proteins inhibit viral pathogens independently of immune sensing. Preliminary evidence suggests that sequestration of Sp1 by IFI16 also promotes HIV-1 latency [65], which represents the major obstacle against the cure of HIV/AIDS.

Altogether, many studies mentioned above clearly support that nuclear PHYHIN proteins silence viral promoters independently of their potential sensing functions. For example, lack of crucial components involved in immune signaling, such as IRF3, ASC, STING, and cGAS, failed to relieve HSV-1 gene expression and infectious virus, unlike the reduction of IFI16 total protein levels [63,68]. Lack of IFN production from HPV-transduced cells [82] and HEK293T co-transfected with proviral HIV-1 and IFI16 expression constructs [65] further confirmed sensing-independent antiviral activity. Recently, IFI16 has been shown to inhibit replication of Zika (ZIKV) and Chikungunya (CHKV) viruses in primary foreskin fibroblasts [95]. At least in the latter case, the antiviral activity was independent of IFN production [96]. In summary, human IFI16 and presumably also other nuclear PYHIN proteins restrict viral transcription by a variety of non-exclusive mechanisms, including epigenetic modifications, occupation of viral promoters, and sequestration of cellular transcription factors, especially Sp1. The relative importance of these mechanisms in innate antiviral immunity and their potential cell type dependencies clearly warrant further study.

## 5. Viral Counteraction and Exploitation of Human PYHIN Proteins

Viruses are well known for their ability to counteract or evade innate immune defenses that would otherwise be able to control them. In fact, the evolution of specific viral antagonists is perhaps the strongest indicator for the relevance of antiviral factors in vivo. Thus, PYHIN proteins seem to play a relevant role in innate antiviral defense since accumulating evidence suggests that various viral pathogens, such as HSV-1, KSHV, HBV, HIV, and HCMV, evolved strategies to antagonize, evade, or even hijack human PYHIN proteins.

One cellular pathway that is commonly exploited by viral pathogens to counteract antiviral factors is the induction of ubiquitination and subsequent proteasomal degradation [2,3,4]. Viruses might either exploit cellular E3-ubiquitin ligases for their own purposes or encode their own viral E3-ubiquitin ligases [97,98,99,100]. For example, it has been reported that HSV-1 utilizes the proteasomal degradation machinery to prevent restriction by IFI16 and IFIX (Figure 6a) [49,75]. It has been proposed that the immediate early transcription factor ICP0 (infected cell protein 0), ubiquitinates IFI16 via the E3 ligase activity of its RING finger domain to promote its proteasomal degradation [49,101]. However, the role of ICP0 as IFI16 antagonist is controversial. In tumor-derived cell lines, ICP0 seems neither sufficient nor required for IFI16 degradation during HSV-1 infection [71]. Another study reported that a virus expressing only ICP0, but no other immediate-early gene, caused only very slow degradation of IFI16 [72]. It has been observed that ICP0-independent degradation of IFI16 correlated with endoribonuclease activity of the virion host shutoff (vhs) protein [102]. Altogether, ICP0 seems to play the major role for IFI16 degradation in non-cancerous human cell lines, while HSV-1 vhs-mediated IFI16 mRNA destabilization seems dominant in tumor-derived cells [102]. Another herpesvirus, KSHV, has also been reported to induce proteasomal degradation of IFI16 upon reactivation from latency [78]. The precise mechanism is unknown but one or more late viral genes seem responsible for this. Since IFI16 inhibits de novo KSHV infection, its degradation may prevent the re-establishment of viral latency [78].

Recent evidence suggests that HBV may antagonize IFI16 at the transcriptional level. Specifically, it has been reported that IFI16 mRNA and protein expression levels are inversely correlated with the abundance of HBV transcripts in liver tissues of patients affected by chronic hepatitis B [88]. The underlying mechanisms are unknown but HBV may silence the *IFI16* promoter activity by inducing epigenetic modifications, inhibition of stimulatory innate immunity signaling pathways, or synthesis of inhibitory noncoding RNAs (Figure 6b) [88].

Intriguingly, primary HIV-1 strains vary enormously in their susceptibility to inhibition by IFI16 and other PYHIN proteins [65,94]. These differences do not depend on the presence of the accessory genes of HIV-1, which are known to play key roles in antagonizing other restriction factors [2,3,4]. Instead, they seem to involve different dependencies on Sp1 for efficient proviral transcription. Notably, highly prevalent subtype C strains of HIV-1 were least sensitive to inhibition by IFI16 and other nuclear PYHIN proteins [65,94]. Altogether, the results indicate that HIV-1 clade C strains evolved reduced dependency of Sp1 to evade restriction by nuclear PYHIN proteins (Figure 6c). It is long known that HIV-1 subtype C LTRs show distinctive features, such as an additional NF-κB interaction site, from those of other subtypes [103,104], but the precise determinants of IFI16 resistance and reduced Sp1 dependency remain to be determined.

Surprisingly, several studies provided evidence that HCMV may hijack IFI16 for efficient spread. Initially, it has been reported that the viral pp65, a major tegument protein and immune evasion factor [105], interacts with IFI16 and recruits it to the major immediate early promoter (MIEP) to enhance its transcriptional activity [106] (Figure 7). It was later shown that pp65 wraps around the PYD of human PYHIN proteins, preventing their oligomerization and dampening IFI16-mediated immune activation [107], as well as AIM2-inflammasome formation [108]. It has been reported that pp65 cooperates with the HCMV encoded serine/threonine-specific kinase pUL97 to phosphorylate IFI16 and to relocate it for virion incorporation [109]. In strict contrast to the HSV-1 ICP0 protein, pp65 was found to protect IFI16 from proteasomal degradation during HCMV infection [110]. In sum, pp65 may hijack IFI16 to fine-tune viral gene expression at the MIEP and capture it into the nascent virions to enhance viral replication fitness [106] (Figure 7). Whether pp65 also modulates the interaction between IFI16 and Sp1 and whether it also hijacks IFIX and MNDA is not known. HCMV establishes latency in undifferentiated hematopoietic cells, in which MIEP expression is repressed [111]. It has been shown that signaling via the HCMV encoded G-protein-coupled receptor US28 is associated with degradation of MNDA and IFI16 during the establishment of viral latency [112]. Since IFI16 promotes immediate early gene expression it is conceivable that its downmodulation might promote the establishment of a latent infection.

## 6. Conclusions and Future Perspectives

Numerous studies investigated the role of PYHIN proteins, especially AIM2 and IFI16, in sensing foreign DNAs. However, the striking finding that lack of the entire PYHIN locus did not affect viral DNA sensing and immune activation in mice [62] makes it questionable whether immune sensing is indeed the major function of PYHIN proteins. A growing body of evidence supports that at least IFI16, IFIX, and MNDA that are mainly localized in the nucleus inhibit viral transcription by various sensing-independent mechanisms including epigenetic modifications and interference with the transcription factor Sp1. Notably, it is becoming evident that many restriction factors, such as APOBEC3G, Tetherin, and TRIM5α, are also involved in viral immune sensing [113,114]. Thus, it is tempting to speculate that PYHIN proteins might also restrict viral transcription, while promoting innate antiviral immune responses, although it will be important to analyze and dissect their various functions in suitable in vivo models. Many questions regarding the exact antiviral mechanisms of human PYHIN proteins, viral countermeasures, cell-type dependencies, and especially the relevance of these effects in vivo remain to be addressed. In addition, the role of nuclear PYHIN proteins and Sp1 in the establishment and maintenance of latency by HIV-1 and Herpesviruses warrants further study. Finally, the transcription factor Sp1 is important for transcription of numerous viral pathogens as well as for expression of cellular genes involved, e.g., in cell differentiation, apoptosis, immune responses, and response to DNA damage, as well as chromatin remodeling in various cancers [115,116,117]. Thus, it will be interesting to further determine the antiviral spectrum and to clarify whether attenuation of Sp1 function by sequestration or occupation of its binding sites might have detrimental effects on cellular functions or even be beneficial because it might reduce the risk of cancer development.

## Figures and Tables

**Figure 1 viruses-12-01464-f001:**
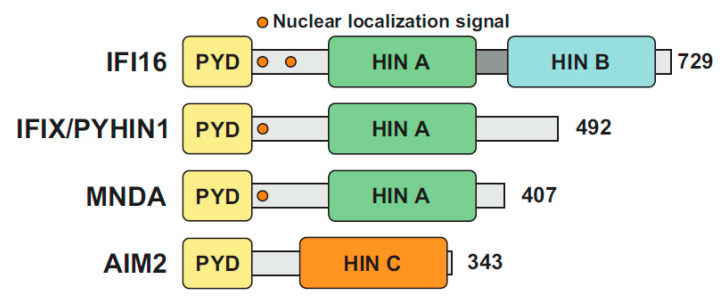
The human pyrin and hematopoietic interferon-inducible nuclear (HIN) domain (PYHIN) protein family. Schematic organization of human PYHIN proteins. Each PYHIN family member possesses an N-terminal pyrin domain (PYD) and one or more HIN domains, classified as HIN A, HIN B and HIN C. With the exception of AIM2, all PYHIN proteins harbor at least one nuclear localization signal (NLS).

**Figure 2 viruses-12-01464-f002:**
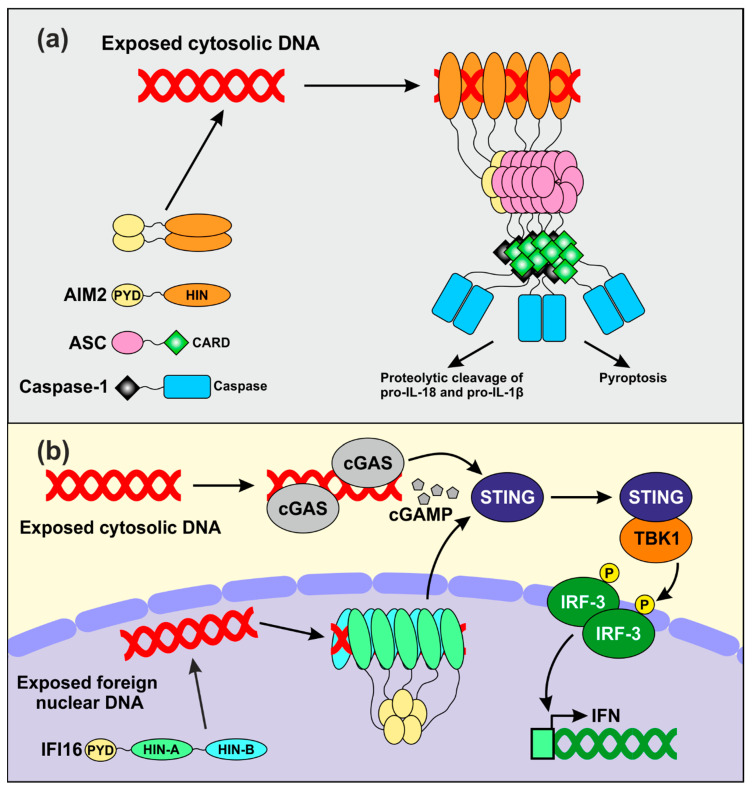
Proposed mechanisms of DNA sensing by AIM2 and IFI16. (**a**) Model for the assembly of the AIM2 inflammasome. AIM2 dimerizes and does not activate any downstream event in the absence of stimulation. Upon binding to exogenous dsDNA, AIM2 assembles filamentous structures stabilized by oligomerization of the PYD. This allows nucleation of ASC filaments which become recruitment platforms for inactive caspase-1. Proteolytic activation of caspase-1 allows subsequent maturation of inactive precursors of inflammatory cytokines and pyroptotic cell death. (**b**) IFI16 and cGAS may cooperate in sensing exogenous DNA. IFI16 senses virus-derived nuclear DNA, while cGAS is responsible for detecting foreign nucleic acids in the cytosol. Both proteins cooperate in activating STING, which in turn stimulates the activity of the TANK binding kinase 1 (TBK1). The interferon regulatory factor 3 (IRF3) is a downstream target of TBK1 which, upon phosphorylation, translocates in the nucleus and activates IFN transcription.

**Figure 3 viruses-12-01464-f003:**
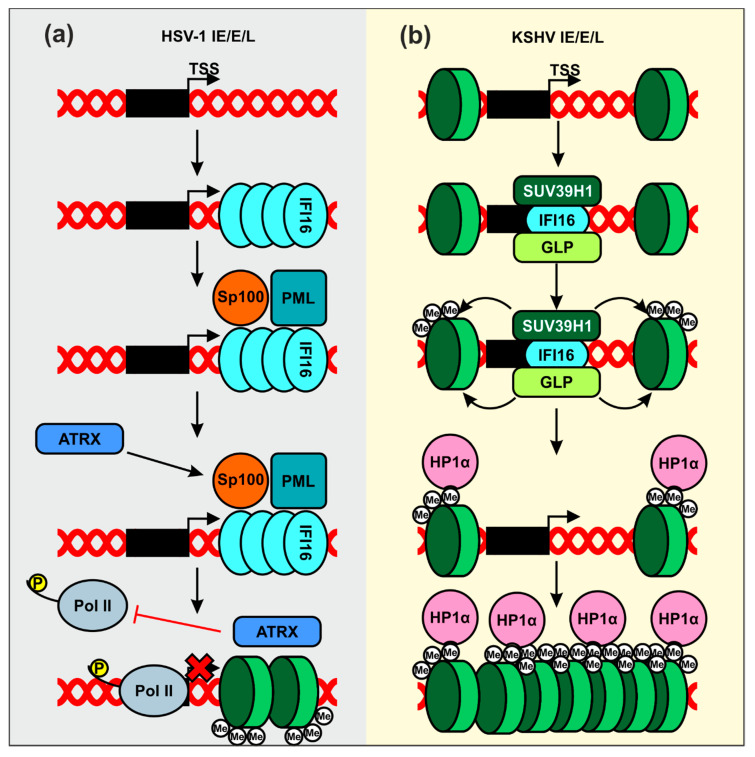
Proposed mechanisms of IFI16-mediated inhibition of herpesviral gene expression by modification of chromatin. (**a**) IFI16 recognizes HSV-1-derived DNA and assembles filamentous structures, which allows recruitment of transcriptional repressors such as ATRX. This reduces the association of elongation-competent RNA Pol II on immediate-early (IE), early (E), and late (L) genes and further leads to chromatinization of viral genes (green discs indicate H3K9). (**b**) IFI16 complexed with SUV39H1 and G9a-like protein (GLP) binds to the viral DNA, and the latter two promote the first methylation event on existing histones. Trimethylated H3K9 is recognized by HP1α, which recruits additional chromatin-modifying enzymes that further compact the viral genes into a repressed state.

**Figure 4 viruses-12-01464-f004:**
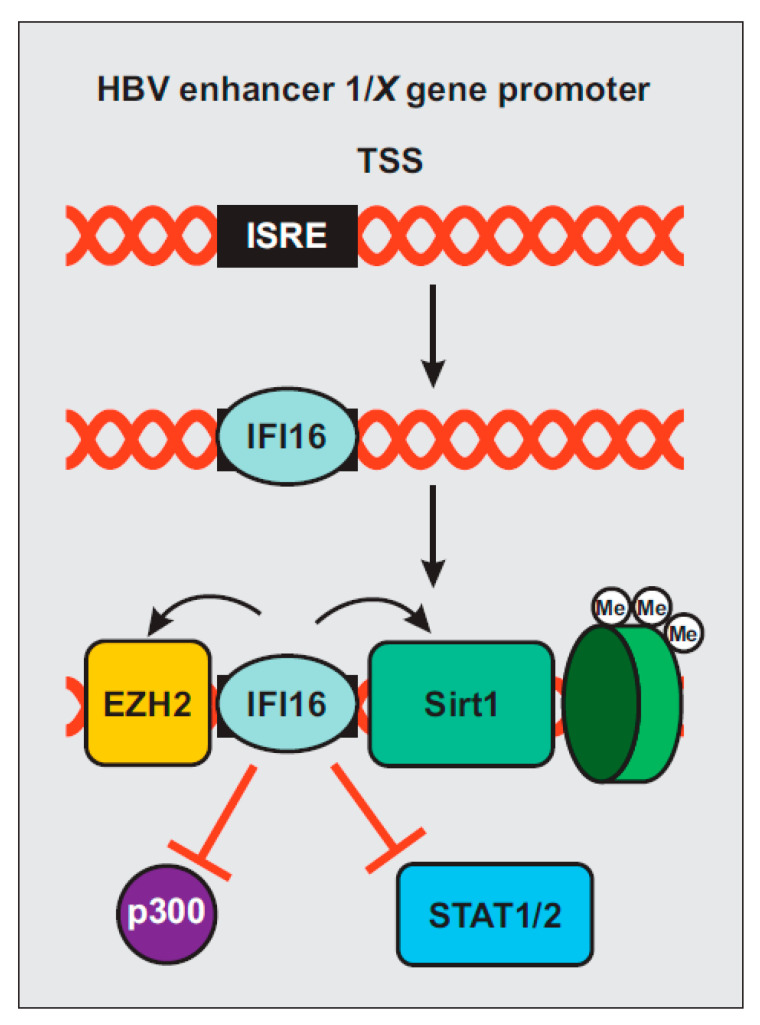
Schematic presentation of hepatitis B virus (HBV) silencing by IFI16. IFI16 selectively binds to an ISRE within the HBV cccDNA. Its recognition facilitates the binding of EZH2, Sirt1 and H3K9me3 while reducing the association of STAT1, STAT2, and p300, leading to epigenetic repression of viral gene expression.

**Figure 5 viruses-12-01464-f005:**
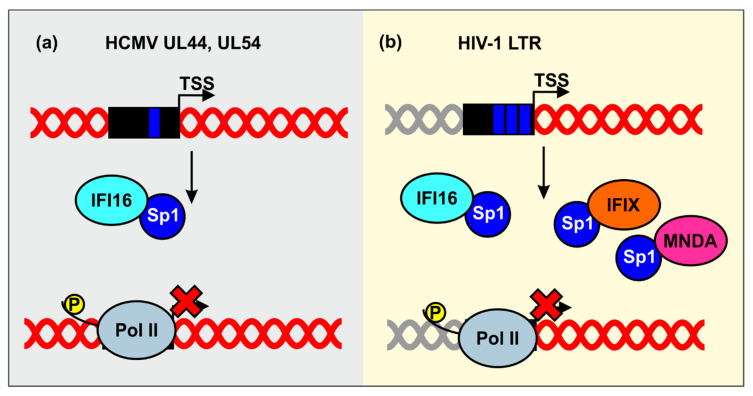
Inhibition of human cytomegalovirus (HCMV) and HIV-1 by sequestration of Sp1. (**a**) IFI16 sequesters the host transcription factor Sp1 to inhibit gene expression of the HCMV UL44 and UL54 gene promoters. (**b**) Nuclear PYHIN proteins reduce the availability of Sp1 to suppress LTR-driven viral gene expression.

**Figure 6 viruses-12-01464-f006:**
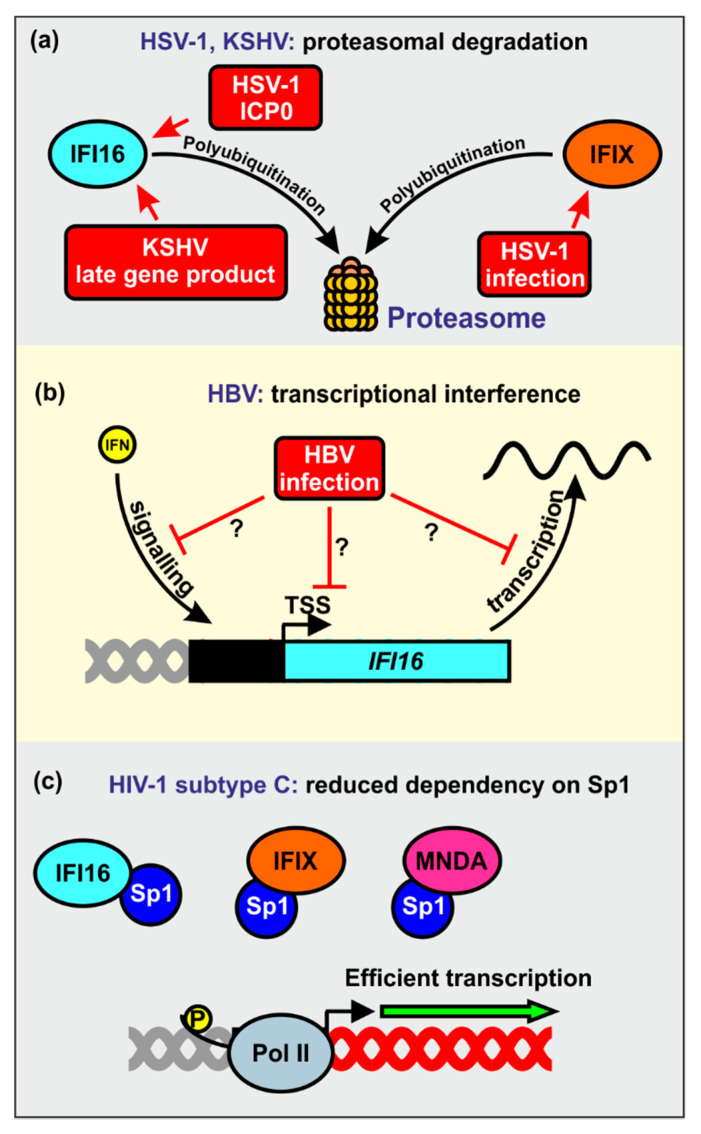
Viral countermeasures against nuclear human PYHIN proteins. (**a**) HSV-1 induces polyubiquitination and proteasomal degradation of IFI16 and IFIX. The viral protein ICP0 is required to efficiently antagonize IFI16. KSHV uses an as yet unidentified late gene product to degrade IFI16 via the proteasome upon reactivation from latency. (**b**) HBV antagonizes IFI16 at the transcriptional level, either by blocking IFN pathways, reducing its basal promoter activity, or by negatively regulating its mRNA. (**c**) HIV-1 subtype C isolates evade PYHIN-mediated restriction via their reduced dependency on Sp1, leading to efficient gene expression even if only low levels of Sp1 are available.

**Figure 7 viruses-12-01464-f007:**
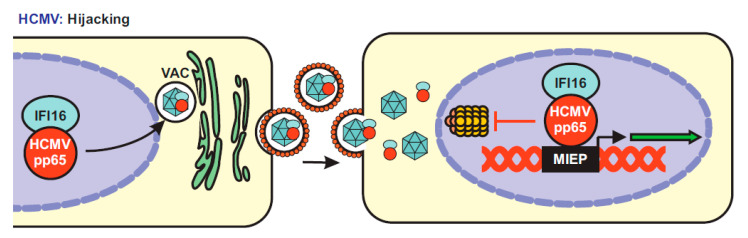
Proposed model of IFI16 utilization by HCMV for efficient transcription. HCMV uses its virion-associated pp65 to hijack IFI16. In the infected cell, pp65 recruits IFI16 to the viral assembly complex (VAC) and incorporates it into the forming virions. Upon a new round of infection, the dimer pp65-IFI16 translocates into the nucleus. pp65 then uses IFI16 to enhance MIEP-driven transcription and protects it from proteasomal degradation.

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
