# Peer review of "Emerging Role of PYHIN Proteins as Antiviral Restriction Factors"

_viruses, 2020, doi:10.3390/v12121464_

Round 1
Reviewer 1 Report
This paper is a very well-written and timely review of PHYN proteins and their emerging role as restriction factors. As someone who is not as familiar with these proteins as I am with other restriction factors, I found the paper informative and very enjoyable to read. The authors have successfully condensed a large amount of research and brought to the forefront the most important points and ideas. Except for some very minor typos that I’ve listed below, I have no corrections.
Line 59 — I think the line “MNDA was the first human PYHIN family member that has been discovered” should read “MNDA was the first human PYHIN family member that was discovered”
Line 61 — “role in” is repeated twice
Line 88 — “ubiquitine” should be “ubiquitin”
Line 90 — “upredulation” should be “upregulation”
Line 118 — The following sentence is confusing:
“In addition, it has been reported that IFI16 promotes activation of the cGAS-STING pathway upon sensing of exogenous DNA in primary macrophages and skin keratinocytes rather than directly acting as DNA sensor.”
I think the authors mean that IFI16 promotes activation after the DNA is detected by cGAS-STING? However, I could be wrong. I think the sentence needs to be clarified.
Figure 3 — I think it would be useful to define IE/E/L in the figure legend. I found the definition in the text, but for those of us not used to herpes viruses, it would be useful not to have to look for it.
Reviewer 2 Report
This is comprehensive review of the PYHIN family of proteins including the protein AIM2, which is a double stranded DNA cytosolic sensor. The rest of PYHIN proteins exert their activities in the nucleus by suppressing viral transcription rather than by sensing viral DNA. Although the review describes in detail this family of proteins, it would be great to give more information on the evidence use to conclude that these proteins block viruses. i.e. some experimental detail (use of KOs versus overexpression, type of cells), the level of the block(10 fold etc), and some information at which stage the viruses are blocked.
- Although AIM2 DNA sensing is described in great length, it would be great to include a virus example, in which AIM2 senses this particular virus? Does this example exist?(line 105)
- For IFI16 Herpes-derived DNAs as target are described, which Herpes viruses where tested? Also it would be good to know some details about these experiments since they suggested a role for IFI16 as a restriction factor (line 115). Level of the block (10 fold, 100 fold?)
- HSV-1 is restricted by IFI16 (line 150), please describe conditions of experiment and level of the block (10 fold, 100 fold?). Histone acetylation may be an indirect mechanism of restriction, please clarify this for the reader. Which is the main mechanism of virus inhibition histone acetylation or ND10 recruitment? Fold of inhibition of each mechanism?
- Similarly, for HBV, HPVs, and HCMV, Zika and others, please describe the level of the block and a few details on the experiments.
